# Revealing the Roles of the JAZ Family in Defense Signaling and the Agarwood Formation Process in *Aquilaria sinensis*

**DOI:** 10.3390/ijms24129872

**Published:** 2023-06-08

**Authors:** Yimian Ma, Jiadong Ran, Guoqiong Li, Mengchen Wang, Chengmin Yang, Xin Wen, Xin Geng, Liping Zhang, Yuan Li, Zheng Zhang

**Affiliations:** 1National Engineering Laboratory for Breeding of Endangered Medicinal Materials, Institute of Medicinal Plant Development, Chinese Academy of Medical Sciences & Peking Union Medical College, Beijing 100193, China; ymma@implad.ac.cn (Y.M.); jdran_0126@163.com (J.R.); 17585563589@163.com (G.L.); mengchenwang98@163.com (M.W.); cmyang@implad.ac.cn (C.Y.); wenxin9979@163.com (X.W.); gx_0117@126.com (X.G.); lpzhang@implad.ac.cn (L.Z.); 2Institute of Plant Protection, Chinese Academy of Agricultural Sciences, Beijing 100193, China

**Keywords:** jasmonate, sesquiterpene, JAZ protein, WRKY transcription factor, expression pattern, protein interaction, phylogenetic analysis

## Abstract

Jasmonate ZIM-domain family proteins (JAZs) are repressors in the signaling cascades triggered by jasmonates (JAs). It has been proposed that JAs play essential roles in the sesquiterpene induction and agarwood formation processes in *Aquilaria sinensis*. However, the specific roles of JAZs in *A. sinensis* remain elusive. This study employed various methods, including phylogenetic analysis, real-time quantitative PCR, transcriptomic sequencing, yeast two-hybrid assay, and pull-down assay, to characterize *A. sinensis* JAZ family members and explore their correlations with WRKY transcription factors. The bioinformatic analysis revealed twelve putative AsJAZ proteins in five groups and sixty-four putative AsWRKY transcription factors in three groups. The *AsJAZ* and *AsWRKY* genes exhibited various tissue-specific or hormone-induced expression patterns. Some *AsJAZ* and *AsWRKY* genes were highly expressed in agarwood or significantly induced by methyl jasmonate in suspension cells. Potential relationships were proposed between AsJAZ4 and several AsWRKY transcription factors. The interaction between AsJAZ4 and AsWRKY75n was confirmed by yeast two-hybrid and pull-down assays. This study characterized the JAZ family members in *A. sinensis* and proposed a model of the function of the AsJAZ4/WRKY75n complex. This will advance our understanding of the roles of the AsJAZ proteins and their regulatory pathways.

## 1. Introduction

Jasmonates (JAs) are essential phytohormones for plant development and survival [1]. Jasmonate ZIM-domain family proteins (JAZs), belonging to the plant-specific TIFY family, are repressors in the signaling cascades triggered by JAs [2,3]. JAZs are involved in JA-mediated and wound-induced signaling pathways and play central roles in the crosstalk between JAs and other phytohormones, such as gibberellic acid, ethylene, and abscisic acid [4,5,6,7,8,9]. JAZs have been identified in many plant species, such as *Arabidopsis thaliana* [10,11,12], *Oryza sativa* [13], *Triticum aestivum* [14,15], *Solanum lycopersicum* [16], *Camellia sinensis* [17,18], and *Vitis vinifera* [19]. They regulate plant defense reactions with specific roles in the signaling pathways of JA-mediated biological processes [20].

Various signaling models involving JAZ have been proposed in *Arabidopsis*. Specifically, it is hypothesized that, in the absence of the active forms of JAs, JAZs can bind to transcription factors and recruit co-repressors to repress gene transcription. In response to pathogen attack or wounds, newly synthesized JAs will facilitate the interaction between JAZ and the SCF complex, leading to the degradation of JAZ and alleviating the repression of JAZ-bound transcription factors [1,21,22,23]. The functions of JAZs can be attributed to two highly conserved sequence regions: the ZIM domain (containing the TIFY motif) and the Jas domain. The ZIM domain and its associated TIFY motif can mediate homo- and heteromeric interactions between most *Arabidopsis* JAZs [24,25]. It can also mediate the interaction between JAZ and the adaptor protein Novel Interactor of JAZ (NINJA), which connects to the co-repressor TOPLESS (TPL) and TPL-related proteins (TPRs) [26,27]. The Jas domain plays a critical role in the jasmonoyl isoleucine-dependent interaction with COI1 to destabilize the repressor [21,28,29]. The functional diversification of JAZ proteins has been established by alternative splicing, and together with the ability of JAZ proteins to homo- and heterodimerize, this has provided pathways for the enhancement of combinatorial diversity and versatility in gene regulation by JAs [25]. Furthermore, the JAZs have overlapping but distinct functions upon interaction with different transcription factors. Since the discovery of JAZ proteins, more than one dozen JAZ-interacting transcription factors have been identified. The best characterized JAZ-interacting transcription factors are in the subgroup of IIIe bHLH proteins, which includes MYC2, MYC3, MYC4, and MYC5 [30,31,32]. However, over the last decade researchers have discovered that WRKY transcription factors are also important JAZ targets, and several JAZ-WRKY modules have been characterized in several plant species. In *Arabidopsis*, Jiang et al. found that WRKY57 interacts with JAZ4/8 and the AUX/IAA protein IAA29, and functions as a node of convergence for JA- and auxin-mediated signaling in JA-induced leaf senescence [33]. Yan et al. discovered that the novel complex JAV1-JAZ8-WRKY51 (JJW) controls JA biosynthesis to defend against insect attack [34]. Chen et al. demonstrated that WRKY75 interacts with several JAZs, such as JAZ8, and proposed that WRKY75 functions as a critical component of the JA-mediated signaling pathways to positively regulate *Arabidopsis* defense responses to necrotrophic pathogens [35]. In *Glycine max*, it was proposed that GmWRKY40 functions as a positive regulator in the response of soybean plants to *Phytophthora sojae* through interactions with JAZ proteins [36]. In *Camelia*, the CsJAZ family members were predicted to be regulated by numerous transcription factors, including ERF, MYB, bHLH, NAC, TCP, AP2, and WRKY family members [18]. WRKY transcription factors can also regulate JAZ expression at the transcription level, but limited studies have explored the regulators of this process [37,38].

*Aquilaria sinensis* (Lour.) Spreng is a species in the genus *Aquilaria* (family Thymelaeaceae). It is an endangered plant that is mainly found in the tropical and subtropical regions of China. Agarwood is one of the most valuable natural incenses in the world, and *A. sinensis* is the only plant resource for agarwood in China [39]. After the trunk or branches of *A. sinensis* have been wounded, sesquiterpenes are produced and accumulate near the wound, forming agarwood as fragrant, resin-filled heartwood. Currently, the natural agarwood resources in most countries are endangered. The traditional agarwood-inducing methods, such as physical wounding and fungal infection, require several years or decades; accordingly, such methods are far from meeting the market needs [40]. Over the past several decades, much effort has been devoted to improving agarwood production, but the agarwood induction mechanism remains elusive. It has been suggested that agarwood is induced by the defense reaction of *A. sinensis*, and that JAs play essential roles in the agarwood induction process, whether this process is induced by wounding or pathogens [41]. To date, multiple lines of evidence have directly or indirectly supported this hypothesis. Further, research has demonstrated that heat shock can induce JA production and sesquiterpene accumulation in *A. sinensis* cell suspension cultures and can increase the expression levels of specific genes in the JA signaling pathways [42]. Studies have also demonstrated that JAs regulate the production of defense secondary metabolites in many plant species via JA-responsive transcription factors, including the AP2/ERF, bHLH, MYB, and WRKY transcription factors [43,44,45,46]. In a previous study, we discovered a novel *A. sinensis* terpene synthase ASS1, which can convert farnesyl diphosphate into δ-guaiene, a major sesquiterpene component in *A. sinensis* [47]. We also discovered that the expression of *ASS1* was up-regulated by AsMYC2 but down-regulated by AsWRKY44, and the interaction between AsJAZ1 and AsMYC2 was confirmed by a pull-down assay [48,49]. Together, these studies indicate that the AsJAZ1 and AsMYC2 transcription factors can participate in JA-mediated signaling pathways to regulate the production of sesquiterpenes in *A. sinensis*. However, to date no study has systematically analyzed the AsJAZ family members and the regulatory roles of AsJAZ proteins in relation to AsWRKY transcription factors. Moreover, while a genome-wide analysis of the *A. sinensis* WRKY transcription factor family has been published, it lacks publicly available genome annotation [50,51]. In the current study, based on the recently released high-quality *A. sinensis* genome, 12 putative *AsJAZ* genes in the AsTIFY family and 64 putative *AsWRKY* genes were systematically revealed and characterized. Through transcriptomic and quantitative real-time PCR (qPCR) analysis, the tissue-specific or hormone-induced expression patterns of the *AsJAZ* genes and *AsWRKY* genes were determined. Furthermore, the interaction between AsJAZ4 and AsWRKY75n was validated by in vivo and in vitro experiments. The findings suggest that, in addition to the AsJAZ1/AsMYC2 module, the AsJAZ4/AsWRKY75n module is another module that exists in the JA pathways in *A. sinensis*. The findings of this study contribute to our understanding of the defense signaling pathways of *A*. *sinensis* and the agarwood formation process.

## 2. Results

### 2.1. Phylogenetic Relationships and Major Domains of A. sinensis TIFY Family Members

The plant-specific TIFY family members, including the JAZ family proteins, play essential roles in plant growth and anti-stress processes. The TIFY family proteins are characterized by the TIFY domain (previously known as the ZIM domain), which contains a highly conserved amino acid pattern (TIF[F/Y]XG; pfam06200) [2]. Here, the hidden Markov model (HMM) profile of the TIFY domain (PF06200) was extracted and the HMMER search program was used to screen the *A. sinensis* TIFY family proteins from all the predicated *A. sinensis* proteins in the GigaDB database. The resulting candidate sequences were submitted for further domain analysis. Finally, 19 *A. sinensis* TIFY family members were retrieved. Of these, the mRNA sequence of *AsJAZ4* was validated by cDNA sequencing. The mRNA and genomic sequences of *AsJAZ4* were submitted to Genbank, with accession numbers OK507019 and OK507018, respectively. The coding sequences of the 12 *AsJAZ* genes are presented in Appendix A. Sequence alignment revealed a high similarity between the coding sequences of *AsJAZ4* (375 bp) and *AsJAZ5* (408 bp). The dissimilarity between the coding sequences of *AsJAZ4* and *AsJAZ5* lay in the A/G variant at site 338, and a single-base insertion at site 341 in *AsJAZ4*. This insertion changed the reading frame of *AsJAZ4*, resulting in a shorter coding sequence of *AsJAZ4* compared with that of *AsJAZ5*. According to the phylogenetic relationships with *Arabidopsis* TIFY members, the 19 *A. sinensis* TIFY members were divided into four subfamilies (TIFY, JAZ, PPD, and ZML) (Figure 1). In these subfamilies, the ZML subfamily included two ZIM proteins (AsZIM1 and AsZIM2) and two ZIM-like proteins (AsZML1 and AsZML2). The 12 JAZ subfamily members from *A. sinensis* were further divided into five groups: JAZ6, -7, -8, and -9 were in group JAZ-I; AsJAZ12 was in group JAZ-II; AsJAZ10 was in group JAZ-III; AsJAZ4, -5, and -11 were in group JAZ-IV; and AsJAZ1, -2, and -3 were in group JAZ-V. The major domain or motif architecture of the *A. sinensis* TIFY family members was retrieved by searching against the SMART database; manual modification was performed based on related references [3,52,53] (Figure 2). Among the 19 *A. sinensis* TIFY family members, the JAZ subfamily had 12 members, each containing a TIFY domain and a Jas motif; the PPD subfamily had a single member, which contained a PPD domain, a TIFY domain, and a Jas motif; the ZML subfamily had four members, each containing a TIFY domain, a Jas motif, and a GATA zinc-finger domain; and the two TIFY subfamily members contained only a TIFY domain. A further search was conducted for specific motifs of the AsJAZ proteins. In *Arabidopsis*, AtJAZ8 lacked a canonical degron that included a short conserved ‘LPIAR’ motif; however, it had an LxLxL-type EAR motif that mediated the transcriptional repression of JAs [54,55]. In the present study, like AtJAZ8, each of the AsJAZ4 and AsJAZ5 proteins had a canonical LxLxL-type EAR motif (LELRL) and lacked the canonical degron. In *Arabidopsis*, there exists a cryptic MYC2-interacting domain (CMID) in AtJAZ1 or AtJAZ10, and this is related to their interactions with AtMYC transcription factors [52,56,57]. Through sequence comparison, AsJAZ10 was found to contain an N-terminal “SQKFLDRRR” motif that was highly similar to the “CMID” core sequence “FQKFLDRRR” in AtJAZ10, which suggests the possible presence of interactions between the AsJAZ10 and MYC proteins. The sequence characteristics of all of the TIFY family genes are summarized in Table 1. It can be seen that, with the exclusion of AsJAZ12, all the remaining putative AsJAZ proteins were small alkaline proteins consisting of no more than 400 amino acids. AsJAZ4 consisted of only 124 amino acids, making it the shortest protein among the putative AsJAZ proteins. The *AsJAZ* genes were located in different chromosomes; their chromosome locations are presented in Appendix A. Based on the sequence locations and positions, *AsJAZ7* and *AsJAZ8* differed only in one base pair and both genes were on chromosome 4; this implies that the two genes are probably paralogous genes that originated from duplication.

### 2.2. Locations and Structures of the AsTIFY Genes and Promoter Elements of the AsJAZ Genes

The structural characteristics of the TIFY family genes were analyzed, particularly in relation to the features of the *AsJAZ* genes (Figure 3a). In terms of the number of exons and their arrangement, *AsJAZ6*, -*7*, -*8*, and -*9* were classified as one group (each having five exons), *AsJAZ11* and *AsJAZ12* were classified as one group (each having five exons and a conserved intron in the TIFY domain), *AsJAZ4* and *AsJAZ5* were classified as one group (each having three exons), *AsJAZ10* was classified as one group (with four exons), and *AsJAZ1*, -*2*, and -*3* were classified as one group (each having seven exons). The intron phase is defined as the position of the intron within or between codons (0, 1, and 2). The *AsJAZ* genes had two intron phase patterns: long (102012) for three genes (*AsJAZ1*, -*2*, and -*3*), and short for the others. The intron position was highly conserved in the sequences encoding the core TIFY domains and the Jas motifs. As shown by the sequence alignment results, a conserved phase 0 intron was inserted in eight of the twelve *AsJAZ* genes after the position encoding the 20th amino acid of the TIFY motifs (Figure 3b). In 10 of the 12 JAZ family members, the 27-amino-acid Jas motif was split into 20 N-terminal and 7 C-terminal (X5PY) amino acid segments by an intron (the Jas intron) located in phase 2 of the codon specifying the Arg residue (R20) at position 20 (Figure 3c). These characteristics are analogous to those reported for *Arabidopsis* JAZ genes [58]. Furthermore, the TIFY motifs were conserved for all 12 AsJAZ proteins, but the 12 Jas motifs were separated into two groups. Nine canonical AsJAZs in several plant species contained the consensus Jas motif “SLX2FX2KRX2RX5PY” [59]. However, AsJAZ4, AsJAZ5, and AsJAZ11 contained divergent Jas motifs like those of AtJAZ7 and AtJAZ8 in the TIFY5 clade [54].

The *cis*-acting element in the promoter region potentially suggests a relationship with growth and defensive responses. The 2 kb upstream sequences of the *AsJAZ* translation initiation sites of the *AsJAZ* genes were searched against the PlantCare database, and the *cis*-acting elements located in the promoters were determined (Appendix A). For the hormone-responsive elements, the promoter regions of the *AsJAZ* genes, except for *AsJAZ3*, -*4*, -*5*, and -*11*, contained the core TGACG (CGTCA) motif, which was homologous to the motifs involved in MeJA responsiveness in *Hordeum vulgare*. The conserved ABA-responsive elements (ABRE) were distributed in all gene promoters. The conserved gibberellin-responsive elements (P-box, TATC box, and GARE motif), auxin-responsive element (TGA element), and salicylic acid–responsive element (TCA element) were also distributed in the upstream promoters of the *AsJAZ* genes. An investigation of the transcription factor binding sites showed that the conserved MYC and MYB transcription binding sites were densely distributed in the promoter regions, while the conserved WRKY transcription factor binding sites (W box) were only distributed in the promoter regions of *AsJAZ2*, -*3*, -*4*, -*5*, -*6*, and -*10*. The findings also revealed several conserved tissue-specific elements, such as motifs or elements, probably related to endosperm expression or meristem expression.

### 2.3. Expression Patterns of the AsJAZ Genes

The raw transcriptome data of a five-year-old *A*. *sinensis* tree were downloaded from the NCBI SRA database with the accession number SRP068230 (released in 2016). Since the genome data of the *A*. *sinensis* tree had not been published at the time, Ye et al. used the de novo transcriptome assembly method to analyze the data [50]. Here, the *A*. *sinensis* tree genome released by Ding et al. was used as a reference [39], and the transcriptome data of the five-year-old *A*. *sinensis* tree were re-analyzed. The FPRM values of all of the genes were re-calculated. The tissue-specific expression levels of the *AsJAZ* genes were estimated through the comparison of their FPKM values, and are shown in a heatmap (Figure 4a). The results revealed that, following the formic acid injection of the *A*. *sinensis* tree for 12 months, *AsJAZ1*, *AsJAZ2*, and *AsJAZ12* exhibited relatively higher expression levels in the white wood (B1), the brown agarwood (J3), and the transition part of the trunk (W2). The expression of *AsJAZ12* was highest in the brown agarwood, and its expression level was significantly higher than the others. In contrast, *AsJAZ3*, *AsJAZ7*, *AsJAZ8*, *AsJAZ9*, and *AsJAZ10* had relatively lower expression levels in the B1, J3, and W2 tissues. However, the five genes displayed relatively higher expression levels in the brown agarwood compared to the other tissues. Among the genes, *AsJAZ6* exhibited a very low expression level in the white wood, and the expression levels of *AsJAZ4*, *AsJAZ5*, and *AsJAZ11* were too low to be detected in the various tissues. The expression patterns of the *AsJAZ* genes in the suspension cells upon various treatments for long (24 h) and short (1 or 2 h) durations were also investigated through RNA sequencing (RNA-seq) or quantitative real-time PCR (qPCR) (Figure 4b and Figure 5). The results showed that the expression levels of *AsJAZ3*, *AsJAZ6*, and *AsJAZ11* remained very low in suspension cells. This suggests that the three genes may be redundant or pseudogenes. Most of the other genes exhibited different expression patterns in suspension cells compared to that in the *A*. *sinensis* tree. For example, although *AsJAZ12* exhibited the highest expression in brown agarwood, it did not always exhibit a higher expression level than the others in suspension cells (Figure 4b and Figure 5). Therefore, it appears that *AsJAZ12* may play a more significant role in the *A*. *sinensis* tree than in suspension cells. RNA-seq analysis revealed that, among all *AsJAZ* genes, the combined expression of *AsJAZ4* and *AsJAZ5* (*AsJAZ4/5*) upon MeJA treatment for 24 h exhibited the highest expression level in suspension cells (Figure 4b). *AsJAZ4/5* also showed a very high expression level upon ABA treatment for 24 h. *AsJAZ10* is another member that was actively expressed in the suspension cells, particularly after the MeJA and ABA treatments for 24 h (Figure 4b). A previous study reported that many *AtJAZ* genes were rapidly induced by JA application for 2 h [11]. Thus, we decided to investigate the expression changes in *AsJAZ* genes after short-duration (1 or 2 h) treatments. The qPCR analysis revealed that the expression levels of *AsJAZ4/5*, *AsJAZ7/8*, *AsJAZ9*, and *AsJAZ10* were in a higher range than those of the other *AsJAZ* genes after the various hormone treatments (Figure 5). Under the MeJA treatment for a short duration (1 or 2 h), the expression of *AsJAZ4/5* increased to the highest among the 12 *AsJAZ* genes. After 2 h of MeSA or ABA treatment, *AsJAZ10* had the highest expression level. After 1 or 2 h of ETH treatment, *AsJAZ7/8* had the highest expression level. To analyze the fold changes in the expression levels of each individual gene under the various hormone treatments, the expression data presented in Figure 5 were re-used to construct Appendix A and the number of genes with significant changes in expression was analyzed (Appendix A). As shown in Appendix A, in suspension cells upon several specific short-duration hormone treatments, some of the *AsJAZ* genes were down-regulated. For example, after MeJA or ETH treatment, the expression of *AsJAZ12* was slightly up-regulated, but after MeSA or ABA treatment the expression was down-regulated. Further, *AsJAZ1* and *AsJAZ2* were down-regulated under short-duration MeJA, MeSA, and ABA treatments.

### 2.4. Chromosome Location of AsWRKY Genes and Phylogenetic Analysis of AsWRKY Proteins

In a previous study, Xu et al. performed a genome-wide analysis of the *A*. *sinensis* WRKY family [51]. However, because the genome sequences were not completely assembled at that time, the analysis results indicated that eight of the predicted 70 *WRKY* genes were located on seven contigs rather than on the chromosome. Additionally, the 70 *WRKY* sequences were not publicly available. In the present study, based on the high-quality *A*. *sinensis* genome presented by Ding et al. [39], the sequence characteristics of *A*. *sinensis WRKY* genes and the phylogenetic relationships between *A*. *sinensis* WRKY proteins were re-examined. Using the HMMER search program, 64 AsWRKY protein sequences were acquired. The 64 *AsWRKY* genes were unevenly located on eight chromosomes; chromosome seven contained the greatest number of WRKY genes (15), but only four *AsWRKYs* were located on chromosome five (Appendix A). There were seven, seven, nine, five, six, and ten *AsWRKY* genes on chromosomes one, two, three, four, six, and eight, respectively.

To determine the phylogenetic relationships between the WRKY proteins from *A. sinensis* and *Arabidopsis*, an unrooted phylogenetic tree was constructed. In contrast to previous reports [60,61,62,63], the WRKY proteins from *A. sinensis* were classified into three groups based on their WRKY domain sequences, using the *Arabidopsis* WRKY proteins as references. The group I members were further divided into subgroup IN (mainly including the N-terminal WRKY domain) and subgroup 1C (mainly including the C-terminal WRKY domain). As shown in Figure 6 and Appendix A, group I WRKYs contained ten AsWRKYs, nine of which contained two WRKY domains, with Scaffold70.27 containing only one N-terminal WRKY domain. The group II members were divided into five subgroups—subgroup IIa, subgroup IIb, subgroup IIc, subgroup IId, and subgroup IIe—which contained four, eight, eighteen, six, and ten AsWRKYs, respectively. Group III contained eight AsWRKYs. Based on the phylogenetic relationships, the orthologous pairs of AsWRKYs and AtWRKYs were determined using a relatively strict criterion; that is, the nodes of the phylogenetic tree with bootstrap support values (1000 re-sampling) exceeding 50% were considered as orthologous pairs, like in the analysis method of Ling et al. [64]. Furthermore, through homology analysis it was discovered that the nucleic acid sequence for Scafford369.3 was identical to that of AsWRKY44, which represses the expression of the wound-induced sesquiterpene biosynthetic gene *ASS1* in *Aquilaria sinensis* [49]. The resulting 25 AsWRKYs orthologous to AtWRKYs are shown in Appendix A. To facilitate the analysis, new names were given to the *A*. *sinensis* proteins orthologous to *Arabidopsis* WRKY transcription factors: AsWRKY33n, AsWRKY44n, AsWRKY1n, etc. This provided a distinction from the nomenclature of Xu et al. [51]. Two gene models with the same *Arabidopsis* orthologous protein AsWRKY40 were named AsWRKY40.1 and AsWRKY40.2. The same approach was taken to naming AsWRKY69.1, AsWRKY69.2, AsWRKY22.1, and AsWRKY22.2. New names were not given to Scaffold21.273 and Scaffold3.179, as each had two *Arabidopsis* orthologous proteins. Although the AsWRKY proteins (Scafford70.29 and Scafford9.167) are not the orthologous proteins for AtWRKY51 and AtWRKY75, according to the strict criterion, the two proteins were closely related to AtWRKY51 and AtWRKY75, respectively, so they were named AsWRKY51n and AsWRKY75n. The IDs and names of the AsWRKYs are shown in Appendix A. Because the sequences of the 70 WRKY protein sequences described by Xu et al. [51] are not available now, only the possible Scaffold IDs were given for some of the AsWRKYs of interest by comparison with their phylogenetic groups (Appendix A). Some members of the group II AsWRKYs, especially group IIc, are highly expressed in agarwood [51]. A comparison of the gene phylogenetic positions indicated that the *AsWRKY* genes specifically expressed in agarwood, named *AsWRKY13*, -*38*, -*49*, -*58*, and -*69* by Xu et al. [51], were Scaffold9.167, -8.462, -271.5, -8.405, and -171.22, respectively.

### 2.5. Expression Patterns of AsWRKY Genes

The expression patterns of the *AsWRKY* genes in the suspension cells in response to hormone and H_2_O_2_ treatments (at a final concentration of 100 μM) for 24 h were investigated based on our primary transcriptomic data. The results are presented in Figure 7. The significance of gene expression changes (treatment vs. control) is shown in Appendix A.

The results revealed that, of the *AsWRKY* genes specifically expressed in agarwood, as described by Xu et al. [51], the expression levels of *AsWRKY75n* and *AsWRKY69* were significantly up-regulated after all of the treatments. In contrast, the expression levels of *AsWRKY65n* were decreased to different degrees after the treatments. However, the expression level of *AsWRKY49* was very low and was not induced by the treatments. The expression level of *AsWRKY58* increased after the hormone and H_2_O_2_ treatments.

Previous studies have proposed that some AtWRKYs can be regulated by AtJAZs, such as AtWRKY51, AtWRKY57, AtWRKY75, AtWRKY40, AtWRKY18, and AtWRKY60 [33,34,35,37,38]. In this study, the expression patterns of several *AsWRKY* genes that have encoding proteins orthologous to the AtWRKY proteins, such as *AsWRKY51n*, -*57n*, -*75n*, -*40.1*, -*40.2*, -*18n*, and -*60n*, were investigated in suspension cells through primary transcriptomic analysis. In these *AsWRKY* genes, *AsWRKY75n* has been included in the *AsWRKY* genes specifically expressed in agarwood, and its expression has been analyzed to be increased under all the treatments. For others, the expression of *AsWRKY51n* was a significant up-regulation after the ABA, ETH, JA, and SA treatments, but there were no obvious changes in response to H_2_O_2_ treatment. There were no significant changes in the expression of *AsWRKY57n* after the various treatments. The expression levels of *AsWRKY40.1* and *AsWRKY40.2* were low during normal conditions, but the expression level of *AsWRKY40.1* increased significantly following all treatments. The expression level of *AsWRKY18n* increased after all of the treatments, but the expression level of *AsWRKY60n* decreased after all of the treatments. *AsWRKY44* has been identified as a repressor of *ASS1* expression and can be induced by MeJA [49]. In the current study, the expression of *AsWRKY44* was found to be significantly increased by all of the treatments.

### 2.6. Experimental Validation of the Interaction between AsJAZ4 and AsWRKY75n

It has been proposed that the expression levels of *AtWRKY70*, *AtWRKY18*, *AtWRKY40*, and *AtWRKY60* could be regulated by JAZ repressors [65,66]. Interactions between specific JAZ and WRKY proteins—such as AtJAZ4/8 and AtWRKY57 or AtJAZ8 and WRKY51/75—have been reported [33,34,35]. As homologous proteins often exhibit analogous functions, hypotheses were made regarding the potential for AsJAZ4 or AsJAZ5 (AtJAZ8 protein homolog) to inhibit the activation of specific AsWRKY transcription factors, including, but not limited to, AsWRKY51n, AsWRKY57n, AsWRKY75n, AsWRKY40.1, and AsWRKY40.2. On the basis of the former analysis, the AsWRKY75n was specifically expressed in agarwood and its expression was increased after some hormone treatments, so we chose to analyze the interaction between AsJAZ4 and AsWRKY75n as a first step to reveal the interactions of AsJAZs and AsWRKYs. Using the RNA extracted from the *A. sinensis* calli as the raw material, the coding sequences of AsJAZ4 and AsWRKY75n were acquired through reverse transcription PCR (RT-PCR). Then, the interaction between AsWRKY75n and AsJAZ4 was investigated using the yeast two-hybrid system. The coding sequences of *AsWRKY75n* and *AsJAZ4* were fused to the Gal4 DNA binding domain of the bait vector (BD-AsWRKY75n) and the Gal4 DNA activation domain of the prey vector (AD-AsJAZ4), respectively. Then, these two vectors were co-transformed into yeast. The Gal4 DNA binding domain expressed by pGBKT7 (BD-Empty1) or the Gal4 DNA activation domain expressed by pGADT7 (AD-Empty2) was used as a negative control. As shown in Figure 8a, after the co-transformation of BD-AsWRKY75n and AD-AsJAZ4, the transformant was able to grow on the synthetic dropout medium lacking Leu, Trp, His, and Ade, while the co-transformation of BD-AsWRKY75n and AD-Empty2, or the co-transformation of BD-Empty2 and AD-AsJAZ4, could not produce yeast with the ability to grow on the synthetic dropout medium lacking Leu, Trp, His, and Ade. These results provide in vivo experimental evidence for the interaction between AsWRKY75n and AsJAZ4, and exclude the possibility of the self-activation of these two proteins. To provide additional confirmation of the interaction between AsJAZ4 and AsWRKY75, a Mag-Beads GST Fusion Protein Purification kit was used to perform a pull-down assay. The results revealed that the immobilized AsJAZ4-GST fusion protein was capable of binding to the AsWRKY75n-His fusion protein, whereas the GST protein was not (Figure 8b). Together, the yeast two-hybrid and pull-down assay results suggest that AsJAZ4 can bind to AsWRKY75n in a similar manner to their *Arabidopsis* homologs.

## 3. Discussion

### 3.1. Classification of the AsJAZ Family Members

The current study identified 19 *AsTIFY* genes, including 12 *AsJAZ* genes. Several conserved *cis*-elements were found in the promoter regions of specific *AsJAZ* genes; these are anticipated to be responsive to various phytohormones, such as MeJA, ABA, GA, auxin, and SA. Previous studies have shown that *Arabidopsis* JAZs can be grouped into four clades [24,25]. Nevertheless, Bai et al. suggested that—based on the topology of the phylogenetic tree, clade support values, and manual inspections—JAZ proteins from angiosperms belong to five groups [58]. Here, the AsJAZ proteins were classified into five groups according to the phylogenetic analysis results. This classification is consistent with that proposed by Bai et al. [58]. Gene structure analysis revealed that conserved intron phases exist in the TIFY and Jas motifs of the same classes of *AsJAZ* genes. This result is also consistent with the findings of Bai et al. [58] with regard to TIFY genes found in *Arabidopsis.* In these AsJAZ proteins, AsJAZ10 contained the conserved core motif of the CMID domain and belonged to the JAZ-III group. Since AsJAZ10 is a homolog of AtJAZ10, it was postulated that AsJAZ10 behaves in a similar manner to AtJAZ10 and potentially interacts with MYC transcription factors [52,57]. In phylogenesis, AsJAZ4, -5, and -11 were located in the JAZ-IV clade, with all three containing the non-canonical Jas motif. Specifically, AsJAZ4 and AsJAZ5 possessed the EAR motif in their N-terminal regions, a characteristic that was also shared by AtJAZ8; this mediates the transcription repression of JA responses in *Arabidopsis* [54]. Therefore, it is hypothesized that AsJAZ4 or AsJAZ5 perform an analogous function to AtJAZ8. However, in the same JAZ-IV group AsJAZ11 did not have the N-terminal EAR motif and exhibited weak relationships with AsJAZ4 and AsJAZ5. On the other hand, in terms of the number of exons and their arrangement, *AsJAZ11* was not grouped with *AsJAZ4* and *AsJAZ5* and instead was grouped with *AsJAZ12*. Therefore, *AsJAZ11* may have different functions to *AsJAZ4* or *AsJAZ5*.

### 3.2. Expression Patterns of the AsJAZ Family Members

The tissue-specific expression patterns of *AsJAZ* genes in a five-year-old *A*. *sinensis* tree injected with formic acid for 12 months were investigated [50]. The findings revealed that *AsJAZ1*, *AsJAZ2*, and *AsJAZ12* exhibited relatively higher expression levels in the white wood (B1), brown agarwood (J3), and transition part of the trunk (W2), as compared to the other tissues. However, the three genes exhibited lower expression levels in the suspension cells. The expression levels of the three genes were even down-regulated upon short-duration treatment with some hormones. This suggests that *AsJAZ1*, *AsJAZ2*, and *AsJAZ12* may be specifically expressed in mature tissues. Furthermore, the three genes may be more strongly correlated with the process of sesquiterpene production and agarwood formation in the *A*. *sinensis* tree than other *AsJAZ* genes. Consistent with this speculation, Liao et al. recently reported the cloning and characterization of a gene named *AsJAZ1* (GenBank accession number: KP677281) from *A. sinensis*, which was identical to the *AsJAZ2* gene in this article [67]. It has been demonstrated that AsJAZ2 (or AsJAZ1, in the report of Liao et al.) can interact with the AsMYC2 transcription factor and may be involved in repressing the expression of sesquiterpenes in *Aquilaria* plants. It is speculated that the functions of AsJAZ1, AsJAZ2, or AsJAZ12 would be to repress the over-production of defense molecules, such as sesquiterpenes, so as to avoid the excessive energy consumption of the tree. As for the investigation of the expression patterns of the *AsJAZ* genes in suspension cells upon the various hormone treatments, *AsJAZ4/5*, *AsJAZ7/8*, *AsJAZ9*, and *AsJAZ10* exhibited higher expression ranges. It is suggested that the functions of *AsJAZ4/5*, *AsJAZ7/8*, *AsJAZ9*, and *AsJAZ10* would be to repress the excessive cell defense response under various treatments. The expression of *AsJAZ4/5* in the suspension cells after short-duration MeJA treatment was significantly higher than the other genes under the various hormone treatments. This suggests that *AsJAZ4/5* plays important roles in the JA-mediated signaling pathways of *A*. *sinensis* suspension cells. Together, these findings provide important guidance for future in-depth analysis of the functions of the 12 *AsJAZ* genes, particularly in relation to their functions in the JA-mediated defense response and the agarwood formation process.

### 3.3. Classification and Expression Patterns of the AsWRKY Genes

This study identified 64 AsWRKY transcription factors based on the high-quality *A. sinensis* genome. Upon phylogenetic analysis, the 64 AsWRKYs were divided into three major groups; the group II WRKRs were further divided into five subgroups (IIa, IIb, IIc, IId, and IIe). The group IIa WRKY genes regulate plant disease resistance by participating in disease-resistance signaling pathways. Some of them, such as GaWRKY1 in cotton and AaWRKY1 in Artemisia annua, regulate the biosynthesis of sesquiterpene phytoalexin [68,69,70]. The present study revealed that the *A. sinensis* genome contains four genes belonging to the group IIa WRKY family: AsWRKY18n, AsWRKY40.1, AsWRKY40.2, and AsWRKY60n. These likely have similar functions to AtWRKY18, AtWYKY40, and AtWRKY60, which are also group IIa WRKYs. AsWRKY33n belongs to group I and may have a similar function to AtWRKY33, which is an important contributor to necrotrophic pathogen disease resistance and interacts with sigma factor binding proteins [71,72]. Previous transcriptome analyses have revealed five *AsWRKY* genes specifically expressed in agarwood, i.e., *AsWRKY13*, -*38*, -*49*, -*58*, and -*69* [51]. Sequence comparisons revealed that *AsWRKY75n* was identical to the *AsWRKY13* described by Xu et al. [51]. The transcriptome analysis in the current study indicated that *AsWRKY75n* was significantly induced by ABA, ETH, MeSA, MeJA, and H_2_O_2_ in suspension cells, implying a possible relationship between this gene and the stress response in agarwood. Further, among the *AsWRKY* genes, *AsWRKY51n* and *AsWRKY75n* were significantly induced by short-duration MeJA treatment in suspension cells. This indicates that *AsWRKY75n* plays a possible role in JA-mediated plant defense against insect and necrotrophic pathogens.

### 3.4. Possible Roles of the AsJAZ4/WRKY75n Complex

In a previous study, the AsJAZ1/AsMYC2 complex was found to be involved in the regulation of ASS1 expression [48,67]. Previous studies have also shown that in *Arabidopsis*, the AtJAZ4/AtWRKY57, AtJAZ8/AtWRKY57, AtJAZ8/AtWRKY51, and AtJAZ8/AtWRKY75 complexes play significant roles in JA-mediated defense pathways [33,34,35]. As homologous proteins often display analogous functions, it was speculated that AsJAZ4/5 (AtJAZ8 protein ortholog) may be a repressor of several specific AsWRKY transcription factors. In keeping with our speculation, a yeast two-hybrid experiment showed an interaction between AsJAZ4 (AtJAZ8 ortholog) and AsWRKY75n (AtWRKY75 ortholog) in vivo and a pull-down assay verified this interaction in vitro (Figure 8). Consistent with this, the qPCR and transcriptomic analysis results indicated that the expression levels of AsJAZ4/5 and AsWRKY75n increased after MeJA treatment, which implies that they are potentially involved in the JA-mediated signaling pathways (Figure 5 and Figure 7). Since unique JAZ/transcription factor complexes are key factors in determining the specificity of the JA signaling pathways, it would be important to investigate the specific roles of the AsJAZ4/WRKY75n complex in the JA-mediated defense pathways. In *Arabidopsis*, AtWRKY75 was found to be involved in various biological processes, such as seed germination, leaf senescence, flowering, salt tolerance, and responding to necrotrophic pathogens [35,73,74,75,76]. In recent years, it has been discovered that DcWRKY75 (a homologue of AtWRKY75n) promotes ethylene-induced petal senescence in carnations [77,78]. As a homologue of AtWRKY75, AsWRKY75n may be a key player actively involved in developmental and defense pathways in *A. sinensis*, and AsJAZ4 may be its repressor. Based on this speculation, a simple model depicting the regulatory role of the AsJAZ4/WRKY75n complex in *A. sinensis* suspension cells was developed. As shown in Figure 9, in normal conditions, AsJAZ4 represses the activities of AsWRKY75n and inhibits the gene’s transcription. Upon MeJA treatment, AsJAZ4 will be degraded and AsWRKY75n will be activated to promote the transcription of the defensive genes. In the meantime, the expression level of the *AsJAZ4* gene was increased to produce new AsJAZ proteins. The newly produced AsJAZ4 proteins will inhibit the activities of AsWRKY75n and avoid the injury originated from the overexpression of the defense genes through a feedback mechanism. This model will provide new insights into the defensive signaling of *A. sinensis*.

## 4. Materials and Methods

### 4.1. Bioinformatic Analysis of TIFY Family Members in A. sinensis

The protein sequences of *A*. *sinensis* were downloaded from the GigaDB database [39]. The hidden Markov model (HMM) profiles of the TIFY domain (PF06200) were extracted from the Pfam database (http://Pfam.sanger.ac.uk, accessed on 21 August 2021) and searches with the HMM model were carried out using the hmmsearch program from the HMMER package against the predicted *A*. *sinensis* proteins [79]. The significant hits (E value < 10^10^) were chosen as possible TIFY proteins. All candidate sequences were submitted for domain analysis using the SMART tool (http://smart.embl-heidelberg.de/, accessed on 21 August 2021) for further validation. For the putative TIFY protein family members, the theoretical pI and predicted molecular weight (MW) were acquired using the Expasy portal tool (http://web.expasy.org/compute_pi/, accessed on 21 August 2021). The conserved motif was analyzed using the MEME online server (http://meme-suite.org/tools/meme, accessed on 21 August 2021). The domain and motif analysis results were visualized using the IBS online illustrator (http://ibs.biocuckoo.org/online.php, accessed on 25 August 2021) [80]. Exon–intron gene structures were visualized using the GSDS 2.0 server (http://gsds.gao-lab.org/, accessed on 25 September 2021) [81]. To analyze the *cis*-acting regulatory elements, 2 kb upstream of the *AsJAZ* genes was extracted and searched against the PlantCARE database (http://bioinformatics.psb.ugent.be/webtools/plantcare/html/, accessed on 25 September 2022) [82,83]. The promoter element analysis results were visualized using TBtools (version 1.09876) [84]. The chromosome location maps were constructed using Mapchart (version 2.3) [85].

### 4.2. Bioinformatic Analysis of WRKY Family Members

The HMM profiles of the WRKY domain (PF03106) were extracted from the Pfam database, and HMMER 3.0 was used to perform local HMM searches against the *A*. *sinensis* proteins in the GigaDB database. The significant hits (E value < 10^10^) were chosen, and the proteins were further screened and validated using the SMART tool. The chromosome location map was constructed using Mapchart (version 2.3) [85].

### 4.3. Sequence Alignment and Phylogenetic Analysis

The representative TIFY protein and WRKY protein sequences of *Arabidopsis* were downloaded from the TAIR database (www.arabidopsis.org, accessed on 25 September 2022), with the accession numbers obtained from the descriptions of Vanholme et al. [2] and Rushton et al. [86]. The TIFY phylogenetic tree construction used the full-length TIFY protein sequences, but the WRKY tree used the WRKY conserved domains. The WRKY conserved domains were extracted from the sequences of the *Arabidopsis* and *A*. *sinensis* WRKY proteins using a local perl script. The N-terminal WRKY domain and C-terminal WRKY domain were separately extracted for the two WRKY domains containing proteins. Both the TIFY tree and WRKY tree were constructed using MEGA-X (version 10.2.6) [87]. First, the amino acid sequences were aligned using MUSCLE, and then the alignment files were imported into MEGA-X for the construction of the Maximum Likelihood tree using 1000 bootstrapped replications. The JTT + G + I + F model was used to construct the TIFY tree, and the JTT + G model was used to construct the WRKY tree. Finally, the phylogenetic trees were modified using the online EvolView tool, with only the nodes with a cutoff greater than 50% shown [88].

### 4.4. Plant Sample Preparation

The suspension cells were derived from *Aquilaria sinensis* (Lour.) Spreng calli and prepared based on a previously reported method [89]. Briefly, the fresh young leaves of *A*. *sinensis* were sterilized using 10% sodium hypochlorite solution, cut into small pieces and cultured on Murashige-Skoog (MS) solid culture medium supplemented with Phytagel (Sigma), α-naphthalene acetic acid (NAA) (2.0 mg·L^−1^), and 6-benzylaminopurine (6-BA; 1.0 mg·L^−1^). After the leaves were cultured and transferred about 5 times, energetic and loose calli were generated and used to prepare suspension cells by shaking in liquid MS medium containing NAA (2.0 mg·L^−1^), 6-BA (1.0 mg·L^−1^), and casein hydrolysate (0.5 g·L^−1^). After the suspension cells were sub-cultured 5–6 times, the fresh and energetic suspension cells acquired after growth for one more week were subjected to one of the following treatments at a final concentration of 100 μM: methyl salicylate (MeSA), methyl jasmonate (MeJA), ethephon (ETH), or (±) abscisic acid (ABA). The suspension cells were treated with various hormones for 0 h (control sample without treatment), 1 h or 2 h, then collected by centrifugation and frozen in liquid nitrogen for real-time quantitative PCR (qPCR). The suspension cells were also treated with MeSA, MeJA, ETH, H_2_O_2_, or ABA (100 μM final concentration) for 24 h and the samples were stored in liquid nitrogen for Illumina-based high-throughput sequencing.

### 4.5. Real-Time Quantitative PCR

Total RNA was extracted using the EASYspin Plus Plant RNA Kit (Aidlab, Bejing, China), according to the manufacturer’s instructions. First-strand cDNA synthesis was performed using M-MLV Reverse Transcriptase according to the instructions (Promega, Madison, WI, USA). The real-time qPCR was performed using TransStart Top Green qPCR Supermix (Transgene, Beijing, China) on a Roche LightCycler96 Real-Time System. The 2^−ΔΔCt^ method was used to evaluate the expression levels. Three independent replications were performed for each experiment. The housekeeping gene *AsTUB* was used as the internal control, as described by Gao et al. [90]. All primers were designed using the online tool Primer3Plus (https://www.primer3plus.com/, accessed on 25 September 2022) and are listed in Appendix A. Because *AsJAZ7* and *AsJAZ8* had highly similar sequences, they used the same pair of qPCR primers; *AsJAZ4* and *AsJAZ5* also used the same pair of primers for their highly similar sequences.

### 4.6. Statistical Analysis

Statistically significant differences (* *p* < 0.05 or ** *p* < 0.01) were determined using the IBM SPSS statistics 26 package (International Business Machines Corp., New York, NY, USA). Data were expressed as the mean + standard error of the mean. Two-tailed Student’s *t*-tests were performed to compare two groups from three independent biological experiments.

### 4.7. Transcriptomic Analysis and Heatmap Illustrating

*A. sinensis* suspension cells were treated with 100 μM MeJA, MeSA, ETH, H2O2, or ABA for 24 h. Three replicates were collected by centrifugation and pooled to form a sample for RNA sequencing on the Illumina Novaseq platform. Sequencing libraries were generated using the NEBNext^®^ UltraTM RNA Library Prep Kit for Illumina^®^ (NEB, Ipswich, MA, USA), following the manufacturer’s recommendations. The generated 150 bp paired-end reads were filtered to obtain clean reads and the HISAT2 v2.0.5 program was used for mapping the clean reads against the *A. sinensis* genome [91]. Gene expression levels were estimated by calculating their FPKM (Fragments Per Kilobase of transcript per Million fragments mapped) value using Cufflinks (version 2.2.1) [92]. Genes with FRKMs values of >1 were regarded as expressed. For the significance analysis of gene expression difference, edgeR software (version 3.38.1) was used [93]. Briefly, the read count was normalized and the *p*-value was calculated via the negative binomial distribution model. Then, the false discovery rate or the error detection rate (FDR, or padj) was calculated. Finally, the genes with significantly changed expression levels between each group (treatment vs. CK) were screened out based on |log_2_ FoldChange| > 1 and a padj value of <0.05. The tissue-specific AsJAZ gene expression data of a five-year-old *A. sinensis* tree were acquired by re-analyzing the transcriptome data previously published by Ye et al. [50]. In brief, the raw data of the transcriptome of the five-year-old *A. sinensis* tree injected with formic acid for 12 months (SRP068230) were downloaded and the reads were re-mapped against the genome released by Ding et al. [50] using the HISAT2 v2.0.5 program, and then the FPKM values for the genes were re-calculated using Cufflinks (version 2.2.1) [92]. Finally, the gene expression levels were compared according to the FPKM values, and the heatmap reflecting the gene’s expression patterns was produced using TB-tools [86].

### 4.8. Protein Expression and Pull-Down Assay 

Coding sequences of *AsWRKY75n* and *AsJAZ4* were PCR-amplified from *A. sinensis* calli cDNAs, with the primers shown in Appendix A. The coding sequence of *AsWRKY75n* was inserted into plasmid pET28a (+) behind the His tag sequence and *AsJAZ4* was inserted into plasmid pGEX-4T-1 behind the GST tag sequence by a homologous recombination method. His-fused AsWRKY75n and GST-fused AsJAZ4 were expressed in *E. coli* BL21-CodonPlus (DE3). Suitable inducing conditions for the soluble protein were determined. The His-fused AsWRKY75n protein was expressed in the supernatant under the induction of 0.2 mM IPTG for 4 h at 18 °C, while the GST-fused AsJAZ4 protein was only partially expressed in the supernatant under the induction of 0.2 mM IPTG for 0.5 h at 18 °C. The pull-down assay was performed using the Mag-Beads GST Fusion Protein Purification kit (Sangon, Shanghai, China), according to the manufacturer’s instructions. Briefly, the AsJAZ4-GST protein or the GST protein produced by the empty vector was adsorbed onto magnetic agarose microspheres pretreated with binding buffer (140 mM NaCl, 2.7 mM KCl, 10 mM Na_2_HPO_4_, and 1.8 mM KH_2_PO_4_, pH 7.4). Then, the AsWRKY75n-His protein was incubated with the AsJAZ4-GST protein and magnetic agarose microspheres complex for 30 min. The unbound proteins were removed by washing twice with washing buffer (same composition as the binding buffer). Finally, the proteins were eluted from the magnetic agarose microspheres using the elution buffer (50 mM Tris-HCl, 10 mM reduced glutathione, pH 8.0), separated by SDS-PAGE and checked with Western blot.

### 4.9. Yeast Two-Hybrid

The yeast two-hybrid experiment was performed via the mating protocol described in the Matchmaker Gold Yeast Two-Hybrid user manual (Clontech, Mountain View, CA, USA). Firstly, the *AsWRKY75* coding sequence was fused to the DNA binding domain of the bait vector pGBKT7 to construct the vector pGBKT7-BD-AsWRKY75n via a seamless cloning method using the EasyGeno Single Assembly Cloning Kit (TIANGEN, Beijing, China). Then, the *AsJAZ4* coding sequence was fused to the activation domain of the prey vector pGADT7 to construct the plasmid pGADT7-AD-AsJAZ4 using a similar method. Then, the two constructs were co-transformed into the yeast strain Y2H Gold. The co-transformation of the empty vector pGBKT7 with pGADT7-AD-AsJAZ4, or pGADT7 with pGBKT7-BD-AsWRKY75n, was used as a negative control. Finally, the success of the co-transformation was detected by the ability of the cells to grow on the synthetic dropout medium lacking Leu and Trp, and the interaction between the co-transformed proteins was detected by the ability of the cells to grow on synthetic dropout medium lacking Leu, Trp, His, and Ade. The primers used for the vector construction are listed in Appendix A.

## 5. Conclusions

This study aimed to examine the characteristics of JAZ and WRKY gene family members in *A. sinensis* and their possible regulation roles in defense signaling and agarwood formation. The findings revealed 12 *AsJAZ* genes. The structures, chromosome locations, and expression patterns of these genes were determined. The phylogenetic relationships between the putative AsJAZ proteins and their *Arabidopsis* homologs were elucidated. The expression analysis showed that *AsJAZ1*, -*2*, and -*12* exhibited higher expression levels than the others in mature trees, while *AsJAZ4/5*, *AsJAZ7/8*, *AsJAZ9*, and *AsJAZ10* were expressed at higher levels in suspension cells. This study identified 64 *AsWRKY* genes. Some *AsWRKY* genes appear to be involved in JA-mediated plant defense pathways, based on the analysis of their expression patterns. This is the first systematic study of the *AsJAZ* genes, and provides experimental evidence for an interaction between AsJAZ4 and AsWRKY75n. The present study has revealed important aspects of the specificity and redundancy of AsJAZ proteins and has uncovered vital insights into their regulation modes via interactions with specific transcription factors. These findings may provide an important basis for future studies of the JA-mediated signaling pathways in *A. sinensis* and the agarwood induction mechanism.

## Figures and Tables

**Figure 1 ijms-24-09872-f001:**
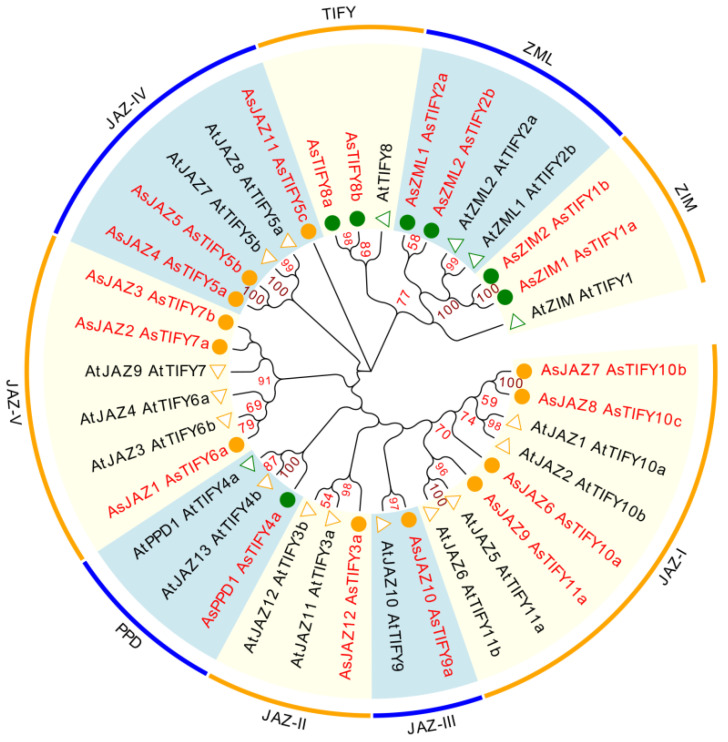
Phylogenetic analysis of TIFY family members from *Arabidopsis* and *A. sinensis*. The phylogenetic tree of TIFY family members from *Arabidopsis* and *A. sinensis* was constructed using MEGA-X software (version 10.2.6) with the Maximum Likelihood (ML) method and 1000 bootstrapped replications. The tree was modified using the online EvolView tool (version 2). The putative *A. sinensis* TIFY proteins are highlighted in red, with orange circles denoting JAZ and green circles denoting other TIFYs. The *Arabidopsis* TIFY proteins are shown with orange triangles (JAZ) or green triangles (other TIFYs).

**Figure 2 ijms-24-09872-f002:**
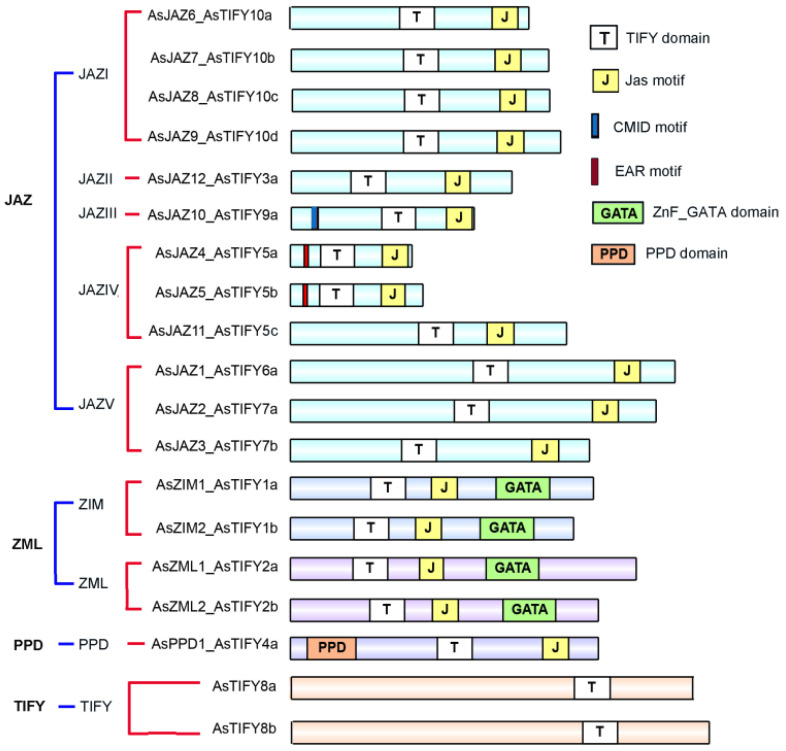
Distribution of the major domains and motifs for the *A*. *sinensis* TIFY family proteins. The AsTIFY proteins are arranged in order of their phylogenetic groups. The rectangles at the bottom represent the complete AsTIFY protein sequences. The domains or motifs were predicated using bio-software SMART (http://smart.embl-heidelberg.de/, accessed on 21 August 2021) or by manual sequence checking; they are presented as small boxes on the rectangles. The proteins, domains, and motifs are differentiated by size and color. In these domains or motifs, AsJAZ4 and AsJAZ5 contain an N-terminal “LELRL” motif (EAR motif). AsJAZ10 contains an N-terminal “XQKFLDRRR” motif (CMID motif) similar to a core motif of the CMID domain of AtJAZ10.

**Figure 3 ijms-24-09872-f003:**
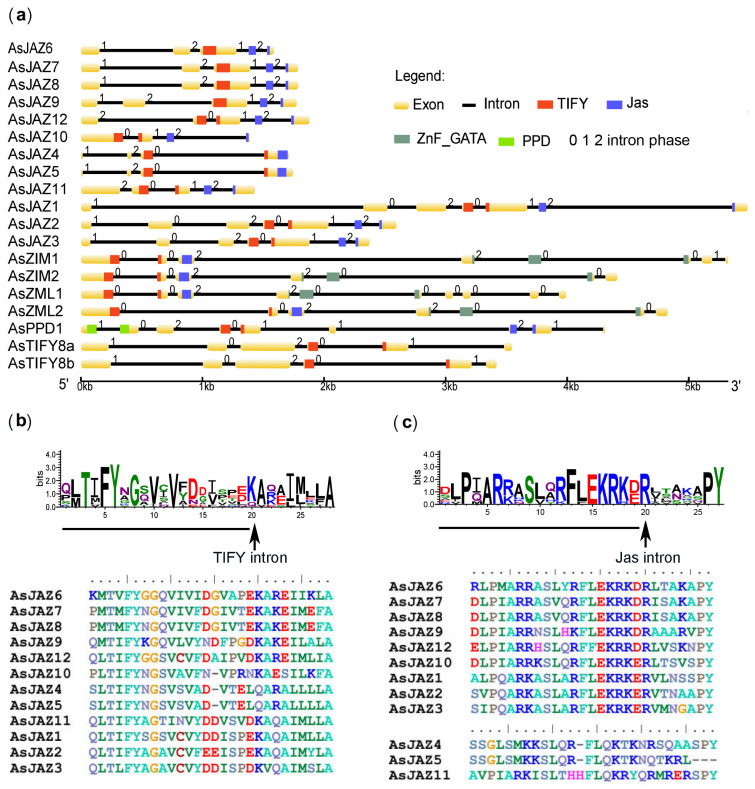
Gene structures and motif logos of the *A. sinensis* TIFY family members. (**a**) Gene structures for the *A. sinensis* TIFY genes. The major protein domains were located in the corresponding gene regions and are shown as different features. (**b**) Consensus sequence logo of the TIFY motif in 12 *A. sinensis* JAZs. The arrow between ‘K’ and ‘A’ of the sequence logo indicates the locations of homologous introns of the TIFY motif for the *AsJAZ* genes, excluding *AsJAZ6*, *AsJAZ7*, *AsJAZ8*, and *AsJAZ9*. The Amino acid alignment revealed that the TIFY motifs of the *A. sinensis* JAZ members AsJAZ6, AsJAZ7, and AsJAZ8 had [T(V/M)FY] consensus sequences that were different from the other members [T(I/L)FY]. (**c**) Consensus sequence logo of the Jas motif of the 12 *A. sinensis* JAZs. The arrow indicates the location of the intron at the Jas motif. The intron was positioned invariably in phase 2 of the codon, specifying the Arg (R) at position 20 of the motif in the sequences of the remaining 10 *AsJAZs*, except for *AsJAZ4* and *AsJAZ5*. The amino acid alignment of the Jas motifs of the *A. sinensis* JAZ members revealed that AsJAZ4, AsJAZ5, and AsJAZ11 had divergent Jas motifs.

**Figure 4 ijms-24-09872-f004:**
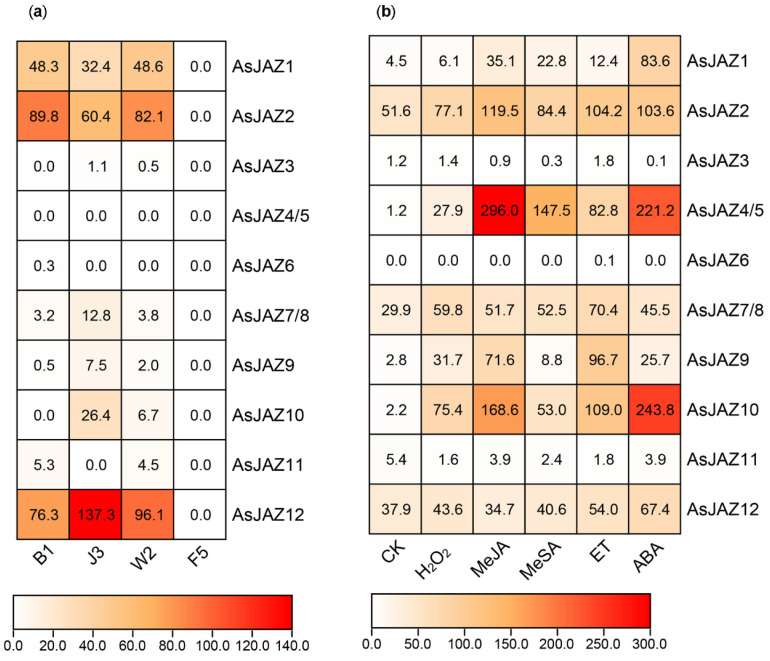
Expression patterns of the *AsJAZ* genes in different *A. sinensis* tissues. (**a**) The heatmap is based on the FPKM values derived from the transcriptomic data published by Ye et al. [50]. The B1, J3, W2, and F5 samples represent different tissues obtained from the trunk of a five-year-old *A. sinensis*. Agarwood was induced after formic acid injection for 12 months. The bark of the *A. sinensis* tree was removed. The white wood in the trunk was the B1 sample. The brown agarwood was defined as the J3 sample. The transition part between the B1 and J3 samples, with a light-brown color, was defined as the W2 sample. The rotten wood in the inner area of the trunk was defined as the F5 sample. (**b**) The heatmap was based on the FPKM values derived from the transcriptomic data of suspension cells upon the various treatments. The treatments included MeSA, MeJA, ETH, H_2_O_2_, or ABA at a final concentration of 100 μM for 24 h. CK is the control sample collected before the treatments. In (**a**,**b**), because the sequences of *AsJAZ7* and *AsJAZ8* only differed in one base pair, the expression FPKM values for the two genes were added up and the total expression values of *AsJAZ7* and *AsJAZ8* are represented by AsJAZ7/8. Similarly, due to the high similarity in the sequences of *AsJAZ4* and *AsJAZ5*, their total expression values are represented by AsJAZ4/5.

**Figure 5 ijms-24-09872-f005:**
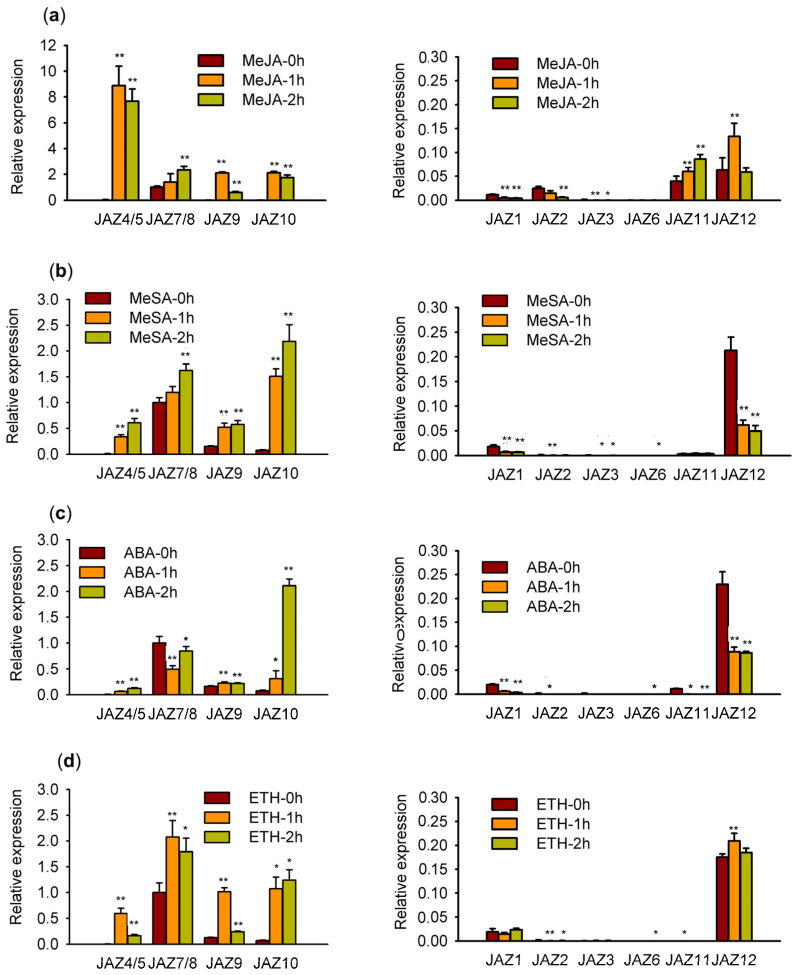
Expression patterns of the *A. sinensis JAZ* genes under the various hormone treatments in suspension cells. The expression data were acquired with real-time qPCR. Because the sequences of *AsJAZ7* and *AsJAZ8* only differed by one base pair, the expression levels of the two genes could not be differentiated and the total expression levels of *AsJAZ7* and *AsJAZ8* are represented by *AsJAZ7/8*. Similarly, because the sequences of *AsJAZ4* and *AsJAZ5* were highly similar, the total expression levels of *AsJAZ4* and *AsJAZ5* are represented by *AsJAZ4/5*. Normal conditions are represented by “0 h”, that is, before treatment, and “1 h” and “2 h” represent 1 h or 2 h after the specific treatment. (**a**–**d**) Indicate the fold changes in the expression levels of the 12 *AsJAZ* genes under the MeJA, MeSA, ABA, or ETH treatment, respectively. The data represent the mean + the standard error of three independent biological replicates. Under each treatment, the relative expression level of *AsJAZ7/8* at 0 h was taken as 1. The *AsJAZ* genes were separated into two groups based on their relative expression level ranges. The significance of the expression difference between the control (0 h) and treatments (1 h or 2 h) for each gene was evaluated using two-tailed Student’s *t*-test, with “*” representing a significant difference at the *p* = 0.05 level and “**” representing a highly significant difference at the 0.01 level.

**Figure 6 ijms-24-09872-f006:**
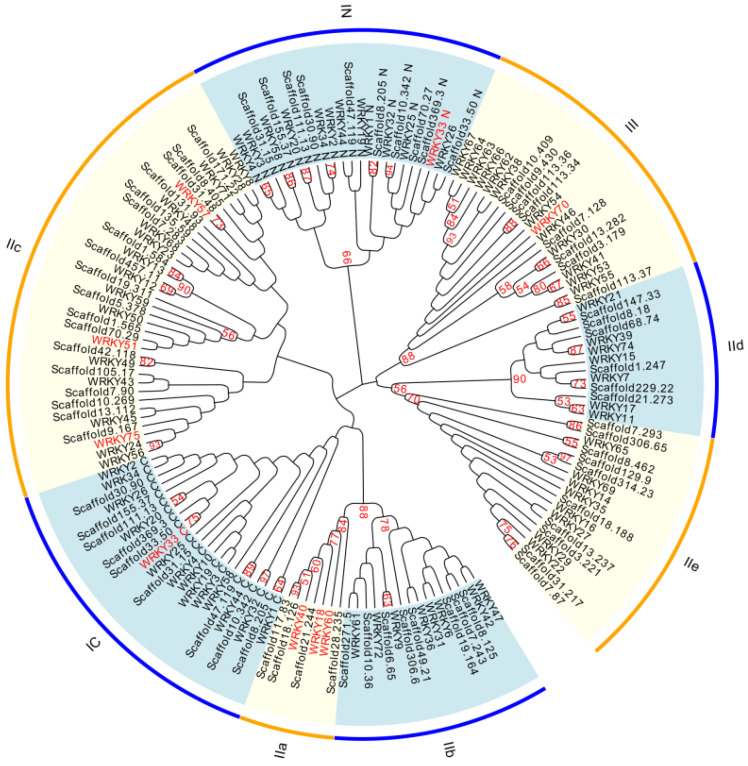
Unrooted phylogenetic tree of *A*. *sinensis* and *A. thaliana* WRKY proteins. The WRKY domains from 64 AsWRKY proteins and 71 AtWRKY proteins were aligned using the build-in MUSCLE program of MEGA X (version 10.2.6) to generate a phylogenetic tree according to the maximum likelihood method with 1000 bootstrap replicates. The AsWRKY proteins are presented by their gene model number, such as “Scaffold28.235”. The AtWRKY proteins are presented by their names with the omission of the prefix “At”. For example, AtWRKY18 is presented as “WRKY18”. For proteins with two WRKY domains, the sequence numbers of the N-terminal and C-terminal domains are marked with “N” and “C”, respectively. The AtWRKY proteins that are probably regulated by AtJAZ proteins are indicated in red. The numbers on the branches of the phylogenetic tree represent the bootstrap values with a cutoff of 50%.

**Figure 7 ijms-24-09872-f007:**
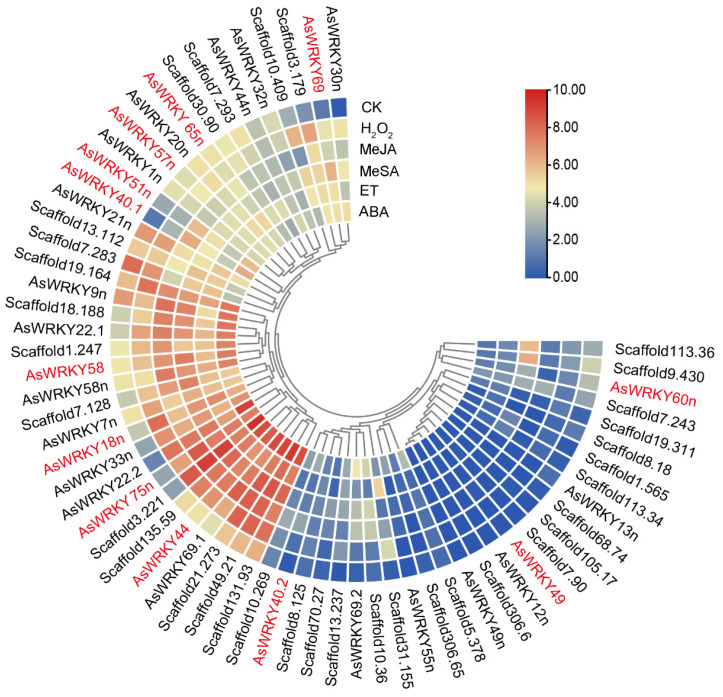
Heat map depicting the *AsWRKY* gene expression patterns under hormone or H_2_O_2_ treatment for 24 h. The final concentration of hormone or H_2_O_2_ treatment was 100 μM. The color changes represent the changes in the log, FRKM values, as indicated in the color key. Blue, yellow, and red represent low, middle, and high expression levels, respectively. *AsWRKY* genes with similar expression patterns are clustered together. The *AsWRKY* genes specifically expressed in agarwood and those encoding proteins with the potential to interact with AsJAZ proteins are written in red.

**Figure 8 ijms-24-09872-f008:**
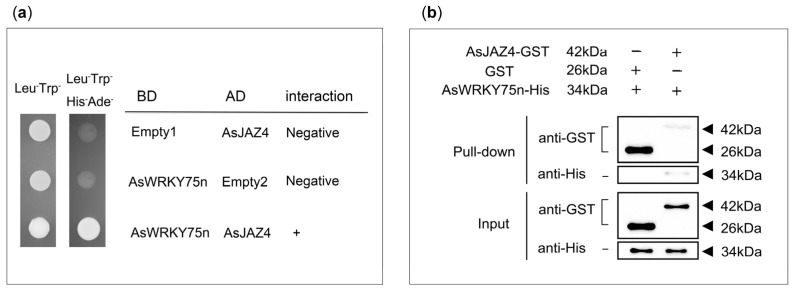
Yeast two-hybrid and pull-down assays. (**a**) Yeast two-hybrid. The interaction was indicated by the ability of cells to grow on synthetic dropout medium lacking Leu/Trp/His/Ade. The negative control co-transformed with an empty vector did not grow on the synthetic dropout medium lacking Leu/Trp/His/Ade. Both the positive and negative transformants grew on the dropout medium lacking Leu/Trp. (**b**) Pull-down assay. AsJAZ4-GST and AsWRKY75n-His represent their corresponding fusion proteins. The GST protein produced by the empty plasmid pGEX-4T-1 was used as a control. After the incubation of immobilized AsJAZ4-GST (GST) and AsWRKY75n-His for 30 min, and the washing off of the unbound proteins, the proteins were eluted from the magnetic agarose microspheres and separated using SDS-PAGE. Then, they were detected using Western blot with GST or His antibodies, as shown in the upper panel (pull-down). The bottom panel (input) shows the loading quantities of the protein samples, as detected by the corresponding antibodies.

**Figure 9 ijms-24-09872-f009:**
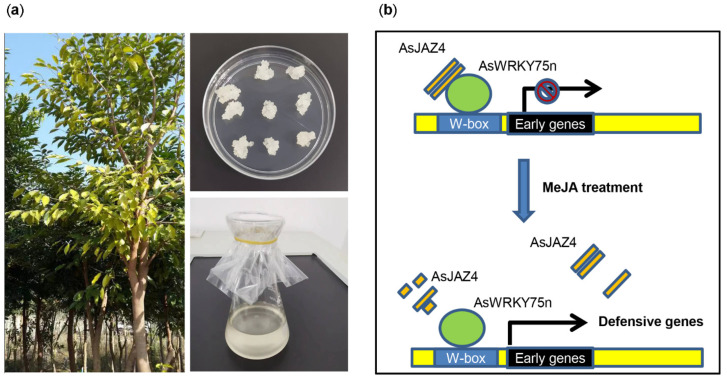
The images of the plant materials and the proposed model of the regulation mechanism of AsJAZ4 to AsWRKY75n. (**a**) The images of plant materials. The tree is the *A. sinensis* plant. The calli of *A. sinensis* are the white irregular cell clusters in the Petri dish and the suspension cells are in the white liquid of the conical flask. (**b**) The proposed model of the regulation mechanism of AsJAZ4 to AsWRKY75n. At normal conditions, low levels of JA-Ile permit the accumulation of the AsJAZ4 dimer (the two orange rectangles) that represses the AsWRKY75 transcription factor (the green ellipse), like other transcription factors, as well. The red prohibitor symbol in the line with an arrow represents the transcription is repressed. After the treatment of MeJA, the increasing levels of JA-Ile promote the degradation of AsJAZ4 and the following transcription of the defense genes. In the meantime, the expression levels of the *AsJAZ4* gene will increase and new AsJAZ4 depressors will be produced.

**Table 1 ijms-24-09872-t001:** The basic characteristics for all putative *AsTIFY* genes.

Name	Synonym	^a^ Chr.	^b^ CDS (bp)	Protein			^f^ ID
				^c^ Length (aa.)	^d^ pI	^e^ MW (Da)	
*AsJAZ1*	*AsTIFY6a*	8 (−)	1161	386	9.39	40,533.48	Scaffold 331.1
*AsJAZ2*	*AsTIFY7a*	4 (−)	1104	367	8.92	38,335.54	Scaffold 5.283
*AsJAZ3*	*AsTIFY7b*	7 (−)	903	300	9.28	31,788.02	Scaffold 134.44
*AsJAZ4*	*AsTIFY5a*	3 (−)	375	124	9.75	13,913.83	Scaffold 262.13 ^g^
*AsJAZ5*	*AsTIFY5b*	3 (+)	408	135	10.01	15,241.81	Scaffold 166.1 ^h^
*AsJAZ6*	*AsTIFY10a*	8 (−)	717	238	9.58	25,777.5	Scaffold 46.130
*AsJAZ7*	*AsTIFY10b*	4 (+)	777	258	9.02	27,571.31	Scaffold 43.66
*AsJAZ8*	*AsTIFY10c*	4 (−)	777	258	8.82	27,518.26	Scaffold 5.466
*AsJAZ9*	*AsTIFY11a*	6 (−)	813	270	7.73	29,085.46	Scaffold 12.166
*AsJAZ10*	*AsTIFY9a*	1 (−)	552	183	8.46	20,269.01	Scaffold 9.133
*AsJAZ11*	*AsTIFY5c*	4 (−)	834	277	9.68	30,599.78	Scaffold 5.102
*AsJAZ12*	*AsTIFY3a*	8 (−)	666	221	6.61	23,444.4	Scaffold 46.99
*AsPPD1*	*AsTIFY4a*	3 (+)	987	328	6.46	35,656.83	Scaffold 22.90
*AsZIM1*	*AsTIFY1a*	2 (+)	915	304	6.17	32,875.36	Scaffold 60.49
*AsZIM2*	*AsTIFY1b*	2 (+)	855	284	6.65	30,628.17	Scaffold 2.660
*AsZML1*	*AsTIFY2a*	2 (+)	1041	346	4.94	38,095.15	Scaffold 60.50
*AsZML2*	*AsTIFY2b*	2 (−)	930	309	5.73	32,843.19	Scaffold 2.738
	*AsTIFY8a*	8 (−)	1212	403	9.66	42,728.3	Scaffold 21.46
	*AsTIFY8b*	4 (−)	1254	417	8.53	44,097.7	Scaffold 23.90

^a^ Chr., the identifier of the chromosomes; (+) or (−) represents plus or minus strand; ^b^ CDS (bp), coding sequence; bp, base pair; ^c^ Length (aa.), the number of amino acids for the predicated peptides; ^d^ pI, isoelectric point; ^e^ MW (Da), molecular weight (Dalton); ^f^ ID, the gene model ID (evm.model. scaffold number); Scaffold 262.13 ^g^ and Scaffold 166.1 ^h^ denote that the two gene models for AsJAZ4 and AsJAZ5 were manually modified based on their conserved EAR motifs and validated by cDNA sequencing.

## Data Availability

The transcriptomic data for *A. sinensis* callus-derived suspension cell samples were submitted to the NCBI SRA database with the accession number SRP323444 (the BioProject number was PRJNA736326). The published transcriptomic data for the five-year-old *A. sinensis* tree injected with formic acid for 12 months was downloaded from the NCBI SRA database with the accession number SRP068230. The gene and peptide sequences are included in the tables in the Supplementary Materials. The genomic and mRNA sequences of *AsJAZ4* have been submitted to Genebank with the accession numbers of OK507018 and OK507019, respectively.

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
