# Peer review of "Revealing the Roles of the JAZ Family in Defense Signaling and the Agarwood Formation Process in Aquilaria sinensis"

_ijms, 2023, doi:10.3390/ijms24129872_

Round 1
Reviewer 1 Report
In this manuscript by Ma et al., the authors analyzed sequences of JAZ family and WRKY family proteins in Aquilaria sinensis, and used RNA-seq to investigate their expression patterns with treatments of stress hormone. Beside that, they focused on AsJAZ4 and AsWRKY75n induced by multiple hormone, detected their interaction and further proposed potential transcriptional regulation of defense and agarwood formation process. Please see the comments below.
Major comments:
For analysis of JAZ gene expression, the authors used hormone treatment for only 1 or 2 hours. However, the RNA-seq was done for 24-hour treatment. Can the authors explain why different timings were chosen? For RNA-seq, the expression patterns of JAZ genes can also be analyzed, but that information was missed in this manuscript.
Minor comments:
Figure_6: "WRKYxx" in phylogenetic tree should be "AtWRKYxx".
Figure_7: Names in red color need to be mentioned in figure legend.
Line_414: "analysis" should be "analyze".
Author Response
1. Major comments:
For analysis of JAZ gene expression, the authors used hormone treatment for only 1 or 2 hours. However, the RNA-seq was done for 24-hour treatment. Can the authors explain why different timings were chosen? For RNA-seq, the expression patterns of JAZ genes can also be analyzed, but that information was missed in this manuscript.
Response: Thank you for your comments. At first, we only analyzed the RNA-seq data, which is acquired by sequencing of the RNAs from suspension cells under 24-hour hormone or H2O2 treatments. But an expert suggested, according to the description of Thines et al. (Nature, 2007, 448, 661–665), many AtJAZ genes are rapidly induced by JA application after two hours, we should study the AsJAZ expression changes in the short-term, such as in 1 h and 2 h. So, we added the qPCR experiment to analysis the expression of AsJAZ genes upon 1 h and 2 h treatments. According to your comment, we think the part of 24-hour treatment should be added. So, we have revised the Figure 4 and added the Heatmap that shows the AsJAZ expression changes upon 24 h treatments. The description of the analysis result of the AsJAZ gene’s expression based on the RNA-seq data (24 h) has also been added. Please see the revised manuscript and the article of Thines et al.
Thines, B., Katsir, L., Melotto, M. et al. JAZ repressor proteins are targets of the SCFCOI1 complex during jasmonate signalling. Nature 448, 661–665 (2007).
2. Minor comments:
Figure_6: "WRKYxx" in phylogenetic tree should be "AtWRKYxx".
Response: We have made revisions to the legend of Figure 6. The description of “WRKYxx” in the Figure legend has been changed as: The AtWRKY proteins are presented by their names with the omission of the prefix “At”. For example, AtWRKY18 is presented as “WRKY18”.
Figure_7: Names in red color need to be mentioned in figure legend.
Response: Have revised.
Line_414: "analysis" should be "analyze".
Response: Have revised.
Reviewer 2 Report
In the Materials and Methods section, subsections 4.1 and 4.8 are repeated.
When describing the treatment of suspension cells with hydrogen peroxide, it is desirable to specify the concentration H2O2.
It is desirable to provide visual images of the plant material under study.
The authors note that genes play a different role in suspension cells and in mature plants. Please describe this in more detail.
Author Response
1. In the Materials and Methods section, subsections 4.1 and 4.8 are repeated.
Response: Have revised
2. When describing the treatment of suspension cells with hydrogen peroxide, it is desirable to specify the concentration H2O2.
Response: Have specified the concentration of H2O2 as 100 μM.
3. It is desirable to provide visual images of the plant material under study.
Response: The images have been added to Figure 9.
4. The authors note that genes play a different role in suspension cells and in mature plants. Please describe this in more detail.
Response: We have revised the section “2.3 Expression patterns of the AsJAZ genes” and “3.2 Expression patterns of the AsJAZ family members”, and described the different of the AsJAZ gene expression patterns in more detail in the two sections.
Reviewer 3 Report
1. Manuscript title: It should be more specific to point out the main theme, particularly about in-depth mechanisms such as the part of JAZs' role in defense signaling, agarwood formation, and interaction with TFs.
2. Keywords: These should be key terms but did not show in the manuscript title, so JAZ, WRKY, and Aquilaria sinensis are not suitable. Also, "pull down" and "yeast two-hybrid" are not good.
3. 4.1 and 4.8 are duplicated.
4. Add a section for statistical analysis.
5. All figure legends should make each figure readable independently, the suggestion is to enrich the description, particularly for Figures 1 and 2.
6. Figure 5: Significant differences between means should be present using the typical format, in which different letters indicate there is a significant difference between means or "*" for significant effect at P=0.05, and "**" at P=0.01.
7. Discussion: Should be enriched, maybe add a conclusive graph for summarizing key molecular mechanisms in this study meanwhile, add related discussion to it.
8. L25-29, L689-693: It should be more specific for the key findings and more clearly point out future directions.
9. The interaction between AsJAZ4 and AsWRKY75n was confirmed: The authors should describe more in the regulating role.
10. To be honest, the current version looks preliminary and may need more experiments to validate more in-depth mechanisms.
Author Response
1. Manuscript title: It should be more specific to point out the main theme, particularly about in-depth mechanisms such as the part of JAZs' role in defense signaling, agarwood formation, and interaction with TFs.
Response: Thank you for your suggestion. The title has been changed from “Comprehensive investigation of JAZ and WRKY family members in Aquilaria sinensis and the discovery of an interaction between AsJAZ4 and AsWRKY75n” to “Revealing the roles of the JAZ family in defense signaling and the agarwood formation process in Aquilaria sinensis”.
2. Keywords: These should be key terms but did not show in the manuscript title, so JAZ, WRKY, and Aquilaria sinensis are not suitable. Also, "pull down" and "yeast two-hybrid" are not good.
Response: Thank you for your suggestion. We have deleted the key words “JAZ; WRKY; yeast two-hybrid; pull-down; Aquilaria sinensis”. The key words have been changed as “jasmonate; sesquiterpene; JAZ protein; WRKY transcription factor; expression pattern; protein interaction; phylogenetic analysis”.
3. 4.1 and 4.8 are duplicated.
Response: Have revised. Please see the revised manuscript.
4. Add a section for statistical analysis.
Response: The section has been added to the “Materials and Methods” part and described as “Statistically significant differences (*P< 0.05 or **P<0.01) were determined using the IBM SPSS statistics 26 package (International Business Machines Corp., New York, USA). Data are expressed as the mean + standard error of the mean. Two-tailed Student’s t-tests were performed to compare two groups from three independent biological experiments.”.
5. All figure legends should make each figure readable independently, the suggestion is to enrich the description, particularly for Figures 1 and 2.
Response: Thank you for your suggestion. We have enriched the description of the figures.
6. Figure 5: Significant differences between means should be present using the typical format, in which different letters indicate there is a significant difference between means or "*" for significant effect at P=0.05, and "**" at P=0.01.
Response: Thank you for your suggestion. We have changed the labels as "*" for significant effect at P=0.05, and "**" at P=0.01.
7. Discussion: Should be enriched, maybe add a conclusive graph for summarizing key molecular mechanisms in this study meanwhile, add related discussion to it.
Response: Thank you for your comments. We have revised the “Discussion” part to describe the key molecular mechanisms in this study in more detail.
8. L25-29, L689-693: It should be more specific for the key findings and more clearly point out future directions.
Response: Thank you for your comments. We have revised the parts according to your opinion.
9. The interaction between AsJAZ4 and AsWRKY75n was confirmed: The authors should describe more in the regulating role.
Response: Thank you for your suggestion. We have added Figure 9 to describe the regulatory process of AsJAZ4 to AsWRKY75n in the last graph of the Discussion part. Please see the revised manuscript.
10. To be honest, the current version looks preliminary and may need more experiments to validate more in-depth mechanisms.
Response: Thank you for your suggestion. Frankly, we have tried to validate more in-depth mechanisms. For example, we have tried to validate the interaction of AsJAZ4 to other AsWRKY transcription factors, such as AsWRKY40, but the interaction was not detected. We have also designed experiments to reveal the genes regulated by AsWRKY75n, such as chromatin immunoprecipitation. I hope the mechanism will be more fully revealed in the future.
Round 2
Reviewer 3 Report
I don't have further questions.